# Near-Infrared Photoimmunotherapy in Brain Tumors—An Unexplored Frontier

**DOI:** 10.3390/ph18050751

**Published:** 2025-05-19

**Authors:** Haruka Yamaguchi, Masayasu Okada, Takuya Otani, Jotaro On, Satoshi Shibuma, Toru Takino, Jun Watanabe, Yoshihiro Tsukamoto, Ryosuke Ogura, Makoto Oishi, Takamasa Suzuki, Akihiro Ishikawa, Hideyuki Sakata, Manabu Natsumeda

**Affiliations:** 1Department of Biochemistry, School of Life Dentistry at Niigata, The Nippon Dental University, Niigata 951-8580, Japan; harukay@ngt.ndu.ac.jp; 2Department of Neurosurgery, Brain Research Institute, Niigata University, Niigata 951-8585, Japan; masayasu_okd@bri.niigata-u.ac.jp (M.O.); jotaro-on_silver@sky.hi-ho.ne.jp (J.O.); satoshikunyo@hotmail.co.jp (S.S.); torutakinons@gmail.com (T.T.); watanabejun1003@gmail.com (J.W.); y-tsukamoto@bri.niigata-u.ac.jp (Y.T.); oguryou@bri.niigata-u.ac.jp (R.O.); mac.oishi@bri.niigata-u.ac.jp (M.O.); 3Department of Brain Tumor Biology, Brain Research Institute, Niigata University, Niigata 951-8585, Japan; 4Near InfraRed Photo-ImmunoTherapy Research Institute, Kansai Medical University, Hirakata, Osaka 573-1010, Japan; otanit@hirakata.kmu.ac.jp; 5Faculty of Engineering, Niigata University, Niigata 950-2181, Japan; takamasa@eng.niigata-u.ac.jp; 6Startup Incubation Center, Shimadzu Corporation, Kyoto 604-8511, Japan; ishikawa@shimadzu.co.jp (A.I.); sakata@shimadzu.co.jp (H.S.); 7Advanced Treatment of Neurological Diseases Branch, Brain Research Institute, Niigata University, Niigata 951-8585, Japan

**Keywords:** near-infrared photoimmunotherapy, IR700, photodynamic therapy, central nervous system, brain tumors, glioma, meningioma, brain metastasis, intraoperative treatment, laser

## Abstract

Near-infrared photoimmunotherapy (NIR-PIT) is a promising cancer treatment that uses near-infrared light to activate a conjugate of a monoclonal antibody (mAb) and a photoactivatable silica phthalocyanine dye (IRDye700DX: IR700). Unlike conventional photodynamic therapy (PDT), NIR-PIT selectively destroys targeted tumor cells while preserving the surrounding normal tissue and providing superior tissue penetration. Recently, NIR-PIT has been approved for the treatment of unresectable recurrent head and neck cancers in Japan. It induces highly selective cancer cell death; therefore, it is expected to be a new curative treatment option for various cancers, including brain tumors. In this review, we compare the principles of NIR-PIT and PDT and discuss the potential applications of NIR-PIT for brain tumors. We selected targetable proteins across various types of brain tumors and devised a strategy to effectively pass the mAb–IR700 conjugate through the blood–brain barrier (BBB), which is a significant challenge for NIR-PIT in treating brain tumors. Innovative approaches for delivering the mAb–IR700 conjugate across the BBB include exosomes, nanoparticle-based systems, and cell-penetrating peptides. Small-molecule compounds, such as affibodies, are anticipated to rapidly accumulate in tumors within intracranial models, and our preliminary experiments demonstrated rapid uptake. NIR-PIT also induces immunogenic cell death and activates the anti-tumor immune response. Overall, NIR-PIT is a promising approach for treating brain tumors. It has the potential to overcome the limitations of conventional therapies and offers new hope to patients with brain tumors.

## 1. Introduction

In brain tumor treatment, balancing the demands of removing as much tumor tissue as possible while preserving normal tissue is challenging. Conventional treatments, such as surgery, radiotherapy, and chemotherapy, each play an important role but also face several limitations. Notably, brain tumors often recur despite the combination of these treatments because of the difficulty of completely eliminating microinvasion at the tumor margins [1,2]. Therefore, photodynamic therapy (PDT) has been used to remove residual cancer cells around the extraction cavity or in unsafe areas of malignant brain tumors [3]. However, photosensitizers used in PDT can accumulate in normal cells and lead to nonspecific phototoxicity, which is not always an acceptable side effect [4,5].

Near-infrared photoimmunotherapy (NIR-PIT) has recently gained attention as an innovative light-based, selective cancer therapy. NIR-PIT is not an improved version of PDT; it offers a fundamentally different approach in terms of its mechanism and clinical potential. NIR-PIT utilizes a conjugate combined with a photoactivated silica phthalocyanine dye (IRDye700DX: IR700) and a monoclonal antibody. It can selectively destroy only targeted tumor cells without damaging the surrounding normal tissue. More importantly, this therapy induces immunogenic cell death in tumor cells and activates an anti-tumor immune response [6].

In recent years, numerous therapies utilizing photosensitizers and activating light have been reported. In these reports, the terms phototherapy and photoimmunotherapy were often used interchangeably without proper distinction, regardless of their mechanisms of action. Consequently, there is no globally standardized definition, which can lead to misunderstanding. In this review, we define and describe NIR-PIT as a therapy that employs IR700 as a photosensitizer and 690 nm near-infrared light to activate the IR700 conjugate, which is currently being clinically applied in Japan.

The selectivity of NIR-PIT is particularly valuable in the brain. Brain tumors, especially glioblastoma, exhibit infiltrative growth patterns and often lack a clear boundary with the normal brain tissue. In such cases, NIR-PIT overcomes the limitations of conventional treatment, offering the potential to completely eliminate these “invisible” tumor cells, representing a new therapeutic strategy to reduce the risk of recurrence. In recent years, extensive molecular-targeting studies have demonstrated the versatility of NIR-PIT. However, there have been few reports of NIR-PIT in brain tumors.

In this review, we compare the principles of PDT and NIR-PIT and also comprehensively explore the application of NIR-PIT in brain tumor treatment. We focus on brain-tumor-specific challenges, including penetrating the blood–brain barrier, antibody delivery systems, and potential molecular protein targets in various brain tumor types for NIR-PIT. In addition, we present preliminary data demonstrating the efficacy of NIR-PIT in brain tumor models. Furthermore, we discuss the anticipated immune responses and adverse effects associated with NIR-PIT in the brain. This review aimed to assess the potential of NIR-PIT as an unexplored frontier in the treatment of brain tumors.

## 2. PDT vs. NIR-PIT

Photodynamic therapy (PDT) originated in 1993, when T. Dougherty et al. from Canada performed PDT to prevent the recurrence of bladder cancer [7]. Targeted treatments have also been conducted for lung cancer and esophageal cancer; however, the photo-sensitizer, porfimer sodium (Photofrin^®^, absorption wavelength 630 nm), demonstrated poor selective accumulation in tumor cells and caused significant skin photo-sensitivity after administration [8]. In contrast, talaporfin sodium (Laserphyrin^®^, absorption wavelength 664 nm), approved for use in 2003 in Japan, is a photosensitizer that addresses the limitations of porfimer sodium [9].

The efficacy of talaporfin sodium in combination with a semiconductive laser emitting light against gliomas has been shown in a series of preclinical studies in Japan [10,11,12]. Subsequently, a two-center, single-arm phase II study testing the efficacy of talaporfin sodium in malignant gliomas was performed, with a 1-year overall survival of 95.5%, a 6-month local control rate of 91%, and, importantly, a relatively safe adverse event profile [13,14]. This study led to the healthcare approval of talaporfin sodium in combination with a semiconductive laser in 2013 in Japan for primary malignant brain tumors. The effectiveness of talaporfin sodium has subsequently been validated in newly diagnosed [14], as well as recurrent [15] glioblastoma at a single high-volume center in Japan.

Talaporfin sodium is intravenously administered 22–26 h before surgery [16]. It accumulates in significantly higher concentrations in cancer cells than in normal cells. The reason for this biodistribution may be the tendency of talaporfin sodium to preferentially combine with low-density lipoprotein (LDL). LDL supplies tissues with the cholesterol necessary to create membranes during cell division. Rapidly dividing cancer cells show increased uptake of LDL, which act as a “transporter” of the photosensitizer to the cancerous tissues [17].

When talaporfin sodium is irradiated with light (664 nm), the energy directs a photosensitizer molecule to the excited triplet state T1 [18]. A hydrogen or electron is transferred, which leads to the formation of free radicals and anion radicals of the photosensitizer and substrate. This process produces reactive oxygen species (ROS) inside the cells, leading to oxidative stress and resulting in the destruction of cancer cells [19,20].

Despite the similarities between PDT and NIR-PIT as cancer therapies that employ photosensitizers and light to activate photochemical reactions, their mechanisms are significantly different.

NIR-PIT utilizes a conjugate synthesized through chemical conjugation of IRDye700DX (IR700) N-hydroxysuccinimide (NHS) ester, a silicon phthalocyanine derivative, to monoclonal antibodies (mAbs) that target the receptors on cancer cells. Approximately three IR700 molecules are bound to each mAb molecule in this conjugate. When the conjugate binds to the receptor, near-infrared light (690 nm) is applied to induce photochemical ligand reactions within the conjugate. Subsequently, the conjugate releases hydrophilic side chains and becomes hydrophobic. This transformation leads the remaining molecules to aggregate and damage the cancer cell membrane, resulting in cell death due to osmotic pressure (Figure 1).

In IR700, the radical anion generated by the electron donor and near-infrared light is protonated by the oxonium ion to form a radical intermediate, which then elongates the Si-O bond and coordinates with a hydroxide ion, cleaving the axial ligand. Because phthalocyanine has a planar structure rich in π electrons, when the axial ligand is cleaved, the molecules stack due to π–π interactions, forming aggregates in aqueous solutions [21,22].

In NIR-PIT, internalization of the conjugate is not necessary to induce cell death. The importance of light irradiation being performed when the IR700 conjugate binds to the receptor on the cell surface should be stressed. Although some conjugates are internalized and generate ROS when exposed to near-infrared light, this is not the primary reason for cell damage [21,23,24].

NIR-PIT demonstrates remarkable selectivity for malignant cell elimination while preserving normal tissues. In Japan, the first NIR-PIT for human use was approved by the Pharmaceuticals and Medical Devices Agency (PMDA) in September 2020. In the U.S., NIR-PIT is currently being evaluated in a phase III clinical trial (ClinicalTrials.gov, NCT identifier: NCT03769506, study number: ASP-1929-301). Clinical validation targeting the epidermal growth factor receptor (EGFR) is particularly notable in head and neck squamous cell carcinoma (HNSCC) [25].

The differences between PDT and NIR-PIT are summarized in Table 1. A critical advantage of NIR-PIT is its significantly higher efficiency, owing to the optimal absorption characteristics of IR700. Talaporfin sodium used in PDT has a very intense absorption peak (Soret band) near 400 nm, a wavelength range not utilized in PDT because of poor tissue penetration, while IR700 has its maximum absorption at 690 nm, which is ideal for deep tissue penetration [26,27]. Generally, the 600–900 nm wavelength region is known as the “optical window”, allowing incident light between these wavelengths to penetrate more deeply into the tissue. This is the reason both NIR-PIT and PDT utilize light within this wavelength range [28]. Although talaporfin sodium exhibits a Q-band between 600 nm and 800 nm, its absorption efficiency is substantially lower than that of IR700 in this therapeutic window. This optimal absorption profile of IR700 at 690 nm allows NIR-PIT to achieve superior tumor-selective targeting, deeper tissue penetration, and enhanced therapeutic potency compared with PDT. Given these differences in photo-physical properties and mechanisms, NIR-PIT should not be viewed merely as a substitute for PDT in brain tumor treatment but rather as a revolutionary therapeutic approach with unprecedented possibilities.

## 3. Therapeutic Potential of NIR-PIT: Consideration of Targetable Surface Antigens Based on Brain Tumor Type

### 3.1. Glioblastoma

The incidence of glioblastoma varies from 3.19 to 4.17 per 100,000 persons-years [31,32], accounting for 14.5% of all central nervous system tumors and 48.6% of malignant brain tumors [33]. Prognosis is still dismal, with a median overall survival of 14.6 to 20.5 months, even after multimodal treatment, including surgery, radiation, and temozolomide [34,35]. In 2013, in Japan, intraoperative photodynamic therapy, involving presurgical intravenous injection of talaporfin sodium in combination with semiconductive laser emitting light, was approved. Intraoperative NIR-PIT is theoretically possible using a similar system, changing the wavelength of the light.

Several studies have been conducted looking at specific cell surface markers in glioblastomas. Chen et al. used the TCGA GBM database to find the top 10 expression-encoded cell surface proteins in glioblastomas, including CD44, CD68, podoplanin, CD163, and TREM2 [36]. Dutiot et al. studied the glioblastoma peptidome to uncover 10 tumor-associated antigens, such as BCAN, FABP7, and NRCAM, to be used in peptide vaccines [37].

Podoplanin is a type-1 transmembrane protein whose function is to bind to C-type lectin-like receptor-2 (CLEC-2) on platelets, inducing platelet aggregation. Physiologically, it is expressed on the lymphatic endothelium [38] with almost no expression in the normal brain or blood vessels [39,40]. We and others have previously shown that podoplanin is highly expressed in IDH-wildtype glioblastomas [41,42,43], whereas in IDH-mutant gliomas, its expression is suppressed by promoter methylation [44]. A preclinical study showed the effectiveness of chimeric antigen receptor (CAR)-T cell therapy targeting podoplanin in an orthotopic glioblastoma model. Podoplanin is also expressed in many tumors, including pleural mesotheliomas [45], lymphangioma [46], meningiomas [47], oral [48] and lung squamous cell carcinomas [49], and head and neck cancers [50], making it an exciting target for NIR-PIT. Preclinical studies showed the effectiveness of the podoplanin antibodies and IR700 conjugate in treating pleural mesothelioma [51] and oral cancer [52]. Kato et al. have produced many podoplanin antibodies, including cancer-specific antibodies that bind to post-translational glycans [53,54,55]. The same group has also developed humanized podoplanin antibodies [56,57].

Epidermal growth factor variant III (EGFRvIII) is an extensively studied surface marker of glioblastoma. It is a product of the EGRF gene with an in-frame deletion of exons 2–7 [58,59,60]. It is a tumor-specific cell surface antigen that constitutively activates the STAT [61] and PI3K–Akt pathways [62] and promotes angiogenesis and tumor growth [63,64,65]. It is known to be expressed mainly in the core of the tumor [66]. EGFRvIII is expressed in 25% of glioblastomas that are IDH-wildtype and EGFR amplified [67]. The peptide vaccine rindopepimut, which targets EGFRvIII, showed promising effects in a phase 2 clinical trial targeting glioblastomas [68], although a subsequent phase 3 trial combining rindopepimut with temozolomide in newly diagnosed glioblastomas expressing EGFRvIII failed to display effects in overall survival [69]. Preclinical studies have shown cytotoxicity of NIR-PIT using the EGFR antibody (Z_EGFR:03115_) in a subcutaneous U87-MGvIII xenograft model [70,71].

Cell adhesion molecule L1 (L1CAM) is a cell surface glycoprotein that promotes tumor migration in proliferation in glioblastomas [72]. L1CAM was found to be expressed in about 20% of both low- and high-grade gliomas [73]. L1CAM is also known to be expressed in supratentorial ependymomas harboring ZFTA fusions [74] and is used as a surrogate marker for this fusion. Other potential targets include GD2 for diffuse midline gliomas [75,76], EphA2 [77], TROP2 [78], IL13Rα2 [79,80], and B7-H3 [81,82,83] for glioblastomas.

### 3.2. Glioma Stem Cells

The literature contains a vast amount of data studying cancer stem cells in glioblastomas. Targeting glioma stem cells is an attractive option, as stem cells are known to be treatment-resistant and have pluripotency and self-renewal properties [84,85,86,87,88,89]. Proposed surface markers of glioma stem cells include CD133 [90,91], CD44 [92], and aldehyde dehydrogenases, such as ALDH1A3 [93]. NIR-PIT studies successfully targeting CD133 [94,95] in glioblastomas have been reported. CD44 is a stem cell marker in mesenchymal-type glioblastomas, the most aggressive type of glioblastoma. CD44–IR700 has been studied in many cancers, such as pancreatic cancer [96,97], triple-negative breast cancer [98], and epithelioid sarcoma [99], but remains to be studied in glioblastomas. ALDH has not been targeted in NIR-PIT studies, but ALDH inhibitors and ALDH peptide-based dendritic cell therapy targeting cancer stem cells [100] have been explored.

### 3.3. Brain Metastasis

Approximately 20% of all cancer patients develop brain metastases and often face a poor prognosis [101]. Against this background, an additional treatment option is required, and NIR-PIT for brain metastases would be a suitable therapeutic approach for these patients.

Lung cancer, breast cancer, and melanoma are the most common primary tumors that metastasize to the brain, accounting for ≥50%, 15–25%, and 5–20% of all brain metastases, respectively [102,103]. Ferguson et al. investigated protein expression on brain metastases from the tumors above to identify potential therapeutic targets. Based on their results and considering the mechanism of NIR-PIT targeting specific proteins on the cancer cell membrane, the targetable proteins are EGFR (expression rate: 66.7%), cMET (31.8%), PDL-1 (22%), PGP (20.1%) for non-small cell lung cancer (NSCLC), HER2 (23.3%) and EGFR (18.8%) for breast cancer, and cMET (36.4%) for melanoma [104]. In particular, HER2 can be an ideal therapeutic target since several studies have shown that HER2 expression is upregulated in brain metastases compared to that in primary breast cancer [105,106]. While PGP is a theoretical target, NIR-PIT targeting PGP seems impractical because it may damage the normal BBB tissues that express PGP [107]. To the best of our knowledge, there are no in vivo NIR-PIT studies using hematogenous brain metastasis models yet. However, NIR-PIT targeting HER2, PD-L1, or PGP has been reported in various extracranial cancers [108,109,110].

Although it remains unclear whether conjugates can cross the BBB, multiple studies have successfully imaged or treated brain metastases using monoclonal antibodies targeting specific proteins [111,112,113]. These reports demonstrated that monoclonal antibodies can cross the blood–brain barrier (BBB) and accumulate in brain metastases, suggesting the feasibility of treating brain metastases using NIR-PIT.

### 3.4. Malignant Meningioma

Meningiomas arise from arachnoid cap cells, forming extra-axial tumors that compress the brain and can eventually invade it. They account for approximately 25–40% of all primary brain tumors [114]. They are generally benign tumors, largely belonging to World Health Organization (WHO) grade 1 tumors. However, recent studies have discovered new prognostic markers, including brain invasion and molecular markers, such as TERT promoter mutation and CDKN2A homozygous deletion [115,116,117]. Integrated diagnosis has raised the percentage of WHO grade 2/3 meningiomas to as high as 38% [115], much higher than that of the ~10% in the pre-molecular era. Notably, grade 3 malignant meningiomas have a poor prognosis, with a 5-year survival rate of approximately 66% and a 10-year survival rate of 14–24% [118,119,120]. Symptomatic or progressively enlarging meningiomas require treatment, with surgical resection being the standard approach. While complete resection, including the dura mater, offers the potential for a cure in benign cases, localization of meningiomas around critical neurovascular structures, such as the dural sinuses, important arteries, cranial nerves, and the skull base, can make curative resection of the tumor by surgery impossible. Additionally, radiotherapy is frequently employed for recurrent, unresectable, or high-grade meningiomas; however, its efficacy remains limited for large tumor volumes and highly aggressive tumors [121]. Residual tumor, even in WHO grade 1 meningiomas, can be problematic. Meningiomas can regrow after radiation treatment. Therefore, there is a need for novel treatments for meningiomas.

Besides podoplanin, somatostatin receptors, especially somatostatin receptor 2 (SSTR-2) and somatostatin receptor 5 (SSTR-5), are frequently expressed in meningiomas and are promising targets for malignant meningioma [122,123,124]. SSTR-2 has been reported to be expressed in 64–100% of meningiomas [122,123,124,125,126,127,128,129], making it an interesting target for NIR-PIT. Importantly, SSTR-2 expression in meningiomas can be pre-operatively assessed by positron emission tomography (PET) using radiolabeled SSTR-2 ligands, such as ^68^Ga-DOTATE [130] or ^68^Ga-DOTATOC [131]. Other somatostatin SSTR2-targeted therapies, including somatostatin analogs [132,133] and peptide receptor radionuclide therapy with 90Y- and 177Lu-DOTATOC [134], are also under investigation.

### 3.5. Functional Pituitary Neuroendocrine Tumor (PitNET)

Neuroendocrine neoplasms (NENs) represent a heterogeneous group of epithelial tumors that arise in various organs, including the gastrointestinal tract and pancreas. They are characterized by morphological and functional features of neuroendocrine differentiation, such as secretory granules, synaptic-like vesicles, and the ability to produce amines and/or peptide hormones [135]. NENs are broadly classified into well-differentiated neuroendocrine tumors (NETs) and poorly differentiated neuroendocrine carcinomas (NECs) [136]. Treatment strategies for NETs depend on the primary site, tumor grade, and disease stage and include surgical resection, somatostatin analogs (SSAs), chemotherapy, and peptide receptor radionuclide therapy (PRRT) [137]. NETs often overexpress specific membrane receptors, particularly SSTRs, enabling both diagnostic imaging and receptor-targeted therapies. PRRT, in particular, has demonstrated efficacy in patients with advanced, progressive, somatostatin-receptor-positive NETs [138]. The clinical success of these receptor-based approaches highlights the therapeutic potential of membrane receptor expression in the management of NETs.

In the 2022 WHO classification (WHO Classification of Endocrine and Neuroendocrine Tumors, 5th Edition), the former term “pituitary adenoma” was replaced with “pituitary neuroendocrine tumor (PitNET)” [139]. While most PitNETs are managed with surgery and/or medical therapy, some PitNETs show treatment resistance or recurrence [140]. Functional PitNETs, like other NETs, frequently overexpress membrane receptors, such as SSTR2, SSTR5, and dopamine receptors, and thus may respond to SSAs [141] and, in selected cases, to PRRT [142]. Given this receptor profile, NIR-PIT, which selectively induces cell death via receptor-targeted antibodies conjugated with photoactivatable dyes, may provide a novel therapeutic option not only for malignant NETs but also for PitNETs with receptor overexpression. However, Volante et al. reported that in poorly differentiated NECs, expression of both SSTR2 and SSTR5 is reduced [143]. Therefore, in cases where PitNET progresses to high-grade pituitary neuroendocrine carcinoma (PitNEC), the effectiveness of SSTR-targeted NIR-PIT may be limited. Careful consideration is warranted in such cases, and the identification of alternative membrane targets may be necessary to develop effective therapies.

## 4. Designing the Optimal Partner for IR700 Conjugate

### 4.1. Challenges in Developing Ideal Conjugates for NIR-PIT in Brain Tumor

The BBB is a highly selective, semipermeable border consisting of brain microvascular endothelial cells connected by tight junctions and covered by pericytes and glial cells. It regulates the passage of molecules between the bloodstream and the brain microenvironment, preventing toxins and pathogens while allowing essential nutrients to pass through [144]. There are several transport mechanisms for substances crossing the BBB, including paracellular pathways, receptor-mediated transport, and cell-penetrating peptides. Generally, only certain gas molecules or highly lipid-soluble, positively charged molecules with a low molecular weight of less than 400 to 500 Da can enter the brain through paracellular pathways [145,146,147]. Approximately 98% of small molecules and nearly all large therapeutic molecules, such as monoclonal antibodies, cannot cross this barrier [146,148].

In primary brain tumors and brain metastases, the BBB is altered to the blood–brain–tumor barrier (BBTB), which is more permeable than the BBB [149]. However, the BBTB is highly heterogeneous, including non-uniform permeability and active efflux of molecules. Tellingen et al. reported that in low-grade gliomas, the BBTB is similar in structure and function to the normal BBB, whereas in high-grade gliomas, increased permeability could be identified in the BBTB. Furthermore, BBTB permeability is high in the bulk of the tumor but low or non-existent in the periphery in glioblastomas [150]. These factors result in inconsistent drug penetration and insufficient therapeutic accumulation in brain tumors [151,152].

Despite these challenging conditions, several antibody therapies for brain tumors, such as bevacizumab and rituximab, are already in clinical use [153]. Theoretically, rituximab, which targets CD20 on the surface of B cells, can be a candidate for the conjugate for NIR-PIT in primary central nervous system lymphoma. Nagaya et al. reported that NIR-PIT using the anti-CD20 rituximab–IR700 conjugate worked highly effectively for B-cell lymphoma [154]. However, studies have shown that a 0.1% to 4.4% concentration of rituximab was detected in cerebrospinal fluid compared with serum [155]. The range of concentration differences is likely due to heterogeneity in the BBTB of each patient. To demonstrate fully effective NIR-PIT, sufficient accumulation of the conjugate in the tumor is necessary. Therefore, even if rituximab stably binds to IR700, it may not be the ideal partner in some cases.

To perform NIR-PIT for brain tumors, the conjugates require properties that overcome these obstacles and effectively cross the BBB or BBTB, enabling sufficient accumulation in tumors. Therefore, various strategies for drug delivery have been explored [156,157].

### 4.2. The Epitope of Antibody for NIR-PIT Is in the Extracellular Domain

The location of an appropriate antibody is critical for the successful application of NIR-PIT. The antibodies used in NIR-PIT are conjugated to a photoactivatable dye, such as IR700 [6]; this section discusses NIR-PIT antibodies from the perspective of antibody-based therapies.

Among the five classes of immunoglobulin (Ig), IgG is predominantly used for therapeutic purposes in immunotherapy, including NIR-PIT. Currently, IgG (cetuximab) is clinically used for NIR-PIT. IgG is a large protein complex composed of two 55 kDa-heavy chains (H chains) and two 25 kDa-light chains (L-chains), linked by disulfide bonds, forming a Y-shaped structure [158]. Because IgG cannot readily pass through intact cell membranes while retaining its native structure and function, therapeutic antibodies are typically directed against extracellular antigens, including cell surface proteins and soluble antigens in the bloodstream [159]. Therapeutically relevant surface antigens include glycoantigens, cell adhesion molecules, receptors, transporters, ion channels, and immune-related markers. In addition, neoantigens classified as tumor-associated antigens may also serve as viable targets in cancer immunotherapy [160].

The distinctive feature of NIR-PIT is that monoclonal antibodies must recognize epitopes that are accessible on the extracellular surface of the plasma membrane. This critical distinction is often overlooked during screening. Failure to select antibodies that bind to accessible regions may result in inadequate photosensitizer delivery and compromised therapeutic efficacy. In other words, as long as they recognize the extracellular domain, not only monoclonal antibodies but also proteins, such as nanobodies, minibodies, and diabodies, enable NIR-PIT therapy.

### 4.3. New Strategies to Deliver IR700 Conjugate to the Brain

To achieve NIR-PIT for brain tumors, new strategies need to be developed to transport sufficient amounts of conjugates while maintaining the ability to bind to targeted receptors on cancer cells. Many different brain delivery platforms for antibodies have been studied, including exosomes, nanoparticle-based systems, and cell-penetrating peptides [161,162].

Chu et al. developed an antibody delivery platform using exosomes that can cross the BBB and effectively deliver bevacizumab, which has been approved to treat glioblastoma. Their animal experiments have illustrated that the introduction of exosomes–bevacizumab into the brain can improve the degeneration of pathological tissues, increase the apoptosis of tumor cells, and significantly extend the survival time [163]. Furthermore, the “Trojan horse” method enables the passage of biomolecules, such as antibodies, through the BBB. Wen et al. developed a two-step Trojan horse to enhance the brain penetration of the monoclonal antibody rituximab. They nano-encapsulated rituximab within a polymer layer that improved BBB penetration, increasing the central nervous system concentrations of the antibody by approximately tenfold compared to naked rituximab [164]. As mentioned above, rituximab is insufficient for use as an NIR-PIT conjugate because of its poor BBB permeability; however, using this method, it may be possible to achieve higher penetration.

While only monoclonal antibodies currently fulfill the role of delivering IR700 in clinical use, recent studies have reported NIR-PIT using alternative carriers, such as peptides, Fabs, and minibodies [165,166,167,168]. Watanabe et al. developed small and bivalent antibody fragments, including the anti-prostate-specific membrane antigen (PSMA) diabody (Db) and minibody (Mb), and compared them to IgG for their effectiveness as photoimmunotherapeutic agents. NIR-PIT with the same molar concentrations of the PSMA-Db–IR700, PSMA-Mb–IR700, and PSMA-IgG–IR700 conjugates showed similar therapeutic effects in vitro and in vivo. They concluded that the use of the PSMA-Db–IR700 conjugate resulted in the shortest time interval between injection and NIR exposure without compromising the therapeutic effects of photoimmunotherapy [169].

In terms of brain tumors, Burley et al. investigated NIR-PIT using an EGFR-specific Affibody conjugated to IR700 for glioblastoma. The Affibody–IR700 conjugate showed specific uptake in vitro and enabled imaging of EGFR expression in the orthotopic brain tumor model. The in vivo study demonstrated the therapeutic efficacy of this conjugate in subcutaneous glioma xenografts [70]. Importantly, it has been reported that there is a significantly higher accumulation of Affibody EGFR–IR800CW than cetuximab–IR680RD in the boundaries of glioma tumors, even though cetuximab has approximately 30 times greater affinity for EGFR than Affibody molecules [70]. Taken together, small molecules, such as Affibody, have the potential to be optimal partners for NIR-PIT in brain tumors.

### 4.4. Preliminary Data Showing Rapid Uptake of Affibody Conjugate in an Intracranial Xenograft Model

Small molecules, such as Affibodies, may efficiently cross the partially disrupted BBB. In the present study, we investigated the time taken for the EGFR Affibody–IR700 conjugate to reach the tumor in an intracranial brain tumor xenograft model. We used a Luminous Quester NX (Shimadzu Corp., Kyoto, Japan) to conduct real-time fluorescence imaging. This in vivo imaging system does not require a dark room and enables simultaneous fluorescence monitoring.

We allowed the growth of RSV-M (EGFR-expressing mouse brain tumor cells) tumors for 2 weeks after cell injection into C3H/HeSlc mice. For real-time imaging, the EGFR Affibody–IR700 conjugate was injected into the tail vein of each mouse. Fluorescence signals were detected in the cranial region in 10 s, reaching plateau levels after 15 s, followed by signal attenuation after 18 s (Figure 2a, Appendix A). One hour after injection of the EGFR Affibody–IR700 conjugate, the brains were extracted, and the fluorescence intensity was evaluated (Figure 2b). After fixation, the brains were sectioned coronally for histological examination. Subsequent hematoxylin and eosin (H&E) staining confirmed the presence of cancer cells in regions exhibiting strong IR700 accumulation (Figure 2c).

Small molecules, such as Affibodies, likely facilitate more rapid tumor accumulation than full antibodies. Current clinical NIR-PIT protocols require IR700 conjugate injection 24 h prior to near-infrared light irradiation. However, our findings suggest that NIR-PIT utilizing small molecular carriers, such as Affibodies, could enable more flexible therapeutic approaches, including intraoperative NIR-PIT administration during surgery.

## 5. Immune Response After NIR-PIT

### 5.1. Mechanisms of NIR-PIT-Induced Immunogenic Cell Death in Brain Tumors

Immunogenic cell death (ICD) is known to trigger an adaptive immune response against cell-related antigens after damage-associated molecular patterns (DAMPs) are released upon cell death [170]. Ogawa et al. reported that when the cell membrane is physically destroyed during NIR-PIT, dying cells release DAMPs, which lead to ICD and the subsequent activation of the antitumor immune system that can address metastatic lesions [171]. It is well recognized that ROS production is essential for the release of DAMPs that lead to ICD [170]. However, when NIR-PIT damages the cancer cell membrane in a unique manner, ROS are not required for the release of DAMPs.

Męczyńska et al. confirmed the release of ROS by NIR-PIT using HER2 Affibody and concluded that ROS production releases ICD [172]. In addition, Nakajima et al. reported that the endocytosed mAb–IR700 conjugate is transported to the lysosome, where ROS induces necrotic cell death after near-infrared light irradiation [173]. Taken together, these results suggest that NIR-PIT induces ICD through two mechanisms: DAMPs released via physical membrane destruction, and ROS generation by IR700 internalized in lysosomes.

Both Ogawa et al. and Męczyńska et al. confirmed key ICD biomarkers, including the relocation of calreticulin, Hsp70, and Hsp90 to the cell surface, along with the rapid release of immunogenic signals, such as ATP and HMGB1, after NIR-PIT. These events induce the maturation of dendritic cells. The matured dendritic cells subsequently lead to CD8+ T-cell activation, enhancing antitumor immunity.

In a glioblastoma mouse model, NIR-PIT activated both CD4+ and CD8+ T cells while significantly increasing the expression of IL-1β and IL-6 cytokines, which can enhance the tumor immune system [71,174]. This suggests that NIR-PIT in the brain causes similar responses to activate the immune system as NIR-PIT in extracranial cancer.

In the in vivo NIR-PIT experiments we conducted using immunodeficient mice (BALB/c-nu), we observed microglia migrating into the brain tumor following near-infrared light irradiation [175]. Although many aspects remain unknown, these findings suggest that NIR-PIT effectively stimulates the immune system in the brain. While NIR-PIT may not eliminate all cancer cells during light irradiation, the remaining cancer cells may be targeted by the activated immune response.

### 5.2. Unique Challenges and Potential of NIR-PIT for Brain Tumors

Conventional cancer immunotherapies activate T cells with cytokines, immune checkpoint inhibitors, such as anti-PD1/PDL1 antibodies, immune suppressive cells, and CART therapy [176,177,178]. These therapies often cause systemic side effects, such as off-target effects, like autoimmune diseases [179]. Immunotherapy for brain tumors is even more difficult because the immune system in the brain has not been fully elucidated.

NIR-PIT only affects the area irradiated with near-infrared light, which is why side effects are unlikely to appear systemically. If NIR-PIT for brain tumors can also activate the immune system in a tumor-specific and partial manner, NIR-PIT could be the most suitable immunotherapy for brain tumors.

Okada et al. reported that Treg-targeted NIR-PIT using anti-CD25 antibodies was effective and that selective removal of Tregs by NIR-PIT activated CD8+ T cells and NK cells, demonstrating an “abscopal” effect that affected distant tumors [180]. Brain Tregs infiltrate the brain during inflammation, such as cerebral infarction. Ito et al. reported that the removal of brain Tregs in cerebral infarction leads to the activation of astrocytes (astrogliosis) and worsening of neurological symptoms. They also found that brain Tregs have characteristic gene expression that is not present in Tregs from other tissues and that brain Tregs interact with astrocytes and contribute to repairing brain damage [181].

Brain immunity has not yet been fully elucidated, and the details of the immune response to NIR-PIT in brain tumors have not been thoroughly examined, even at the basic experimental level. Therefore, considering these differences in Tregs, it may be difficult to control Tregs in the same manner as in extracranial cancers.

Furthermore, recent research has revealed that immune cells, such as monocytes and B cells, are supplied directly to the brain and spinal cord from the bone marrow within the skull. It has also been shown that these cells, which usually remain in the meninges, may infiltrate the brain and spinal cord parenchyma in response to inflammation or injury [182]. These cell infiltrations may occur after NIR-PIT of brain tumors. However, the details of the reaction remain unclear and need to be thoroughly investigated.

## 6. Treating Adverse Events in NIR-PIT for Brain Tumors

When NIR-PIT is applied to HNSCC patients in clinical practice, severe swelling is known to occur as a side effect [25,183]. Therefore, a tracheostomy is performed in patients with a risk of airway obstruction after NIR-PIT for HNSCC during the surgery. The area of swelling is not limited to the tumor site, and although it has been reported that COX2 inhibitors suppress swelling in mouse models after NIR-PIT, the underlying mechanism remains unclear [184]. Experiments conducted in mouse models have demonstrated that NIR-PIT in brain tumors causes swelling [71]. Consequently, the most important aspect of safety in NIR-PIT treatment is the control of transient swelling. This is critical when considering NIR-PIT treatment for brain tumors, as elevation of intracranial pressure can cause herniation. Performing a craniectomy first and then a cranioplasty after the swelling has subsided is possible, although it would involve multiple surgeries. Hinge/floating craniotomy [185] or expansive cranioplasty [186] are reasonable alternatives to decompressive craniectomy, which would mitigate the need for multiple surgeries. Osmotic agents, such as glycerol and mannitol, are regularly used to relieve intracranial pressure [187]. The usage of corticosteroids would also be an effective treatment, although their use may reduce secondary effects brought about by the immune response after NIR-PIT. Finally, bevacizumab, a vascular endothelial growth factor (VEGF)-A monoclonal antibody that produces potent anti-angiogenetic effects while relieving brain edema, is approved for use in glioblastomas. It is known to have corticosteroid-sparing effects [188]. Rigorous monitoring of brain edema after NIR-PIT will be mandatory, and the exact methods to control intracranial pressure remain to be elucidated.

## 7. NIR-PIT for Brain Tumors

As previously mentioned, applying NIR-PIT for brain tumors requires establishing a precise method to control intracranial pressure. If this can be achieved, it may allow for NIR-PIT of brain tumors, similar to the current NIT-PIT for head and neck tumors in Japan. For shallow lesions of tumors, a frontal diffuser is used to directly irradiate light forward, whereas for deep lesions of tumors, a cylindrical diffuser is inserted into the tumor and irradiates light in a cylindrical shape [189,190].

Moreover, given the unique environment of the brain, there may be unique methods to activate conjugates. For instance, employing longer-wavelength near-infrared light to penetrate deeper areas, minimizing scattered light with optical filters to enhance treatment efficiency, using pulse irradiation to activate photosensitizers more effectively, and possibly combining photosensitizers with X-ray-sensitive materials to improve access to deep tumors using X-rays may all be valid methods.

Currently, the most feasible and realistic approach is to use NIR-PIT for margins after tumor removal rather than attempting to treat the entire mass of the brain tumor. To perform NIR-PIT for brain tumors, it is possible to apply the techniques used in PDT, which are currently used in clinical practice.

NIR-PIT demonstrates greater selectivity than conventional treatments and is expected to become a promising new therapeutic approach for brain tumors. This technique may address the challenge of treating brain tumors: it aims to remove as much tumor tissue as possible while minimizing damage to the surrounding normal tissue.

## 8. Conclusions

NIR-PIT represents a revolutionary approach for brain tumor treatment through antibody-guided targeting combined with localized light activation. Although significant challenges remain, including optimal antibody delivery across the BBB, management of edema, and understanding brain-specific immune responses, the potential benefits are substantial.

NIR-PIT shows new therapeutic possibilities and illuminates a path toward an unexplored frontier, offering renewed hope for patients with brain tumors.

## Figures and Tables

**Figure 1 pharmaceuticals-18-00751-f001:**
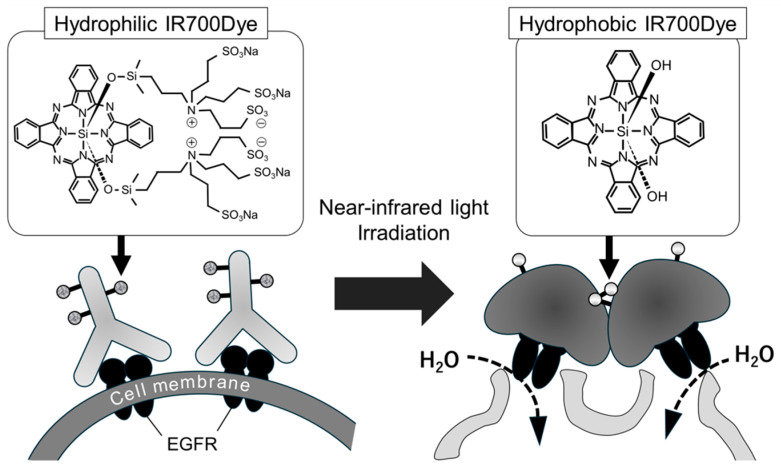
Mechanism of NIR-PIT: NIR-PIT utilizes IRDye700DX–antibody conjugates. When these conjugates bind to cancer cell receptors and are activated by 690 nm near-infrared light, IR700 undergoes photochemical reactions that convert it from hydrophilic to hydrophobic. This transformation leads to molecular aggregation, which disrupts cell membrane integrity and induces cell death. Solid black arrows indicate IR700Dye in each conjugate, while dotted black arrows indicate H_2_O entering the cancer cells.

**Figure 2 pharmaceuticals-18-00751-f002:**
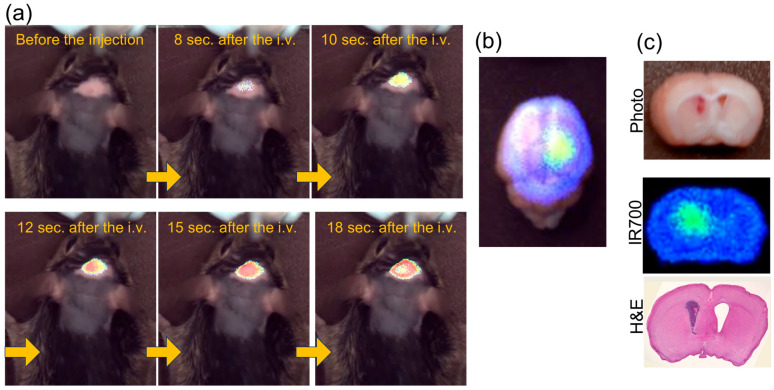
(**a**) The fluorescence intensity of the head increased dramatically immediately after the injection of the EGFR Affibody–IR700Dye conjugate. (**b**) Brain collection and fluorescence imaging were performed 1 h after i.v. injection of the EGFR Affibody–IR700 conjugate. (**c**) Fluorescent images of IR700 and hematoxylin/eosin staining after the brain was fixed overnight. Fluorescence accumulation is observed at the tumor site. The yellow arrows show the progression of the experimental timeline from left to right.

**Table 1 pharmaceuticals-18-00751-t001:** The differences between NIR-PIT and PDT.

Characteristics	NIR-PIT	PDT
Photosensitizer	IR700 conjugated to antibodies	Talaporfin sodium
Excitation wavelength	690 nm	664 nm
Penetration depth	Approximately 2 cm [24]	<10 mm [29,30]
Cell death mechanism	Disruption of cell membranes	Apoptosis via ROS generation
Specificity	High (targets only cells expressing specific antigens)	Moderate (preferential accumulation in tumors)

## Data Availability

Not applicable.

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
