# Peer review of "Near-Infrared Photoimmunotherapy in Brain Tumors—An Unexplored Frontier"

_pharmaceuticals, 2025, doi:10.3390/ph18050751_

Round 1
Reviewer 1 Report
Comments and Suggestions for Authors
Dear Editor-in-Chief of the Journal of Pharmaceuticals
Thank you for allowing me to revise the manuscript “Near-infrared photoimmunotherapy in brain tumors- an unexplored frontier,” which was sublimated to the journal “Pharmaceuticals” with Manuscript ID: Pharmaceuticals - 3589287. The submitted manuscript suits the Pharmaceuticals, and some interesting results were shown. Although there are some scientific errors and some very important parameters that affect Near-infrared photoimmunotherapy (NIR-PIT) and Photodynamic therapy (PDT), they were not addressed in this review article. Based on this, the authors have to consider some requirements. In this regard, I recommend publishing it in the Pharmaceuticals Journal after making major revisions. The following comments are requested:
- I believe the most significant finding should be included in the abstract, along with the study's novelty.
- In the abstract, the author writes the abbreviation IR700 without mentioning the meaning of it. Please mention the mean of IR700.
- The authors mention, “In this review, we compare the principles of PDT and NIR-PIT” in the aim of the review, but they don’t mention this important goal in the abstract.
- The authors mentioned in the manuscript that red light is the light intended by Near-infrared (NIR). Red light falls within the visible light region, which is limited between 350 and 700 nm, while the NIR region includes wavelengths between 700 and 2500 nm. Please revise this scientific point.
- The manuscript neglected the effect of the selected wavelength used to activate the monoclonal antibody and photosensitizer which is one of the most important factors affecting the results of both NIR-PIT and In addition, there are many studies that have developed the type of electromagnetic wave used to activate monoclonal antibodies and photosensitizer by using lasers or X-rays instead of regular light sources and the use of optical filters and the extent to which these filters affect the activation process and results. I hope you take these important factors into account in the test.
Best Wishes
Author Response
Reviewer 1
Comment 1
> I believe the most significant finding should be included in the abstract, along with the study's novelty.
Response:
Thank you for your suggestion. We have revised the abstract to emphasize our most significant findings, including innovative approaches for delivering the mAb-IR700 conjugate across the blood-brain barrier (BBB) and rapid tumor uptake demonstrated by small-molecule compounds in intracranial models.
Comment 2
> In the abstract, the author writes the abbreviation IR700 without mentioning the meaning of it. Please mention the mean of IR700.
Response:
We have added an explanation of IR700 to the abstract. (line 20)
Comment 3
>The authors mention, “In this review, we compare the principles of PDT and NIR-PIT” in the aim of the review, but they don’t mention this important goal in the abstract.
Response:
Thank you for pointing this out. We have revised the abstract to include, “In this review, we compare the principles of PDT and NIR-PIT. " (line 26)
Comment 4
> The authors mentioned in the manuscript that red light is the light intended by Near-infrared (NIR). Red light falls within the visible light region, which is limited between 350 and 700 nm, while the NIR region includes wavelengths between 700 and 2500 nm. Please revise this scientific point.
Response:
You have raised an important point here. In our previous draft, we referred to the laser light used in PDT (664 nm) as 'red light' and the light used in NIR-PIT (690 nm) as 'near-infrared light.' As you have correctly pointed out, this terminology could lead to confusion. Therefore, in Section 2, ‘PDT vs NIR-PIT,' we eliminated the term 'red light' and specified the wavelengths where necessary.
Comment 5
>The manuscript neglected the effect of the selected wavelength used to activate the monoclonal antibody and photosensitizer which is one of the most important factors affecting the results of both NIR-PIT
Response:
Agree. We have added an explanation about the “optical window” for wavelengths in lines 148-151.
Comment 6
> In addition, there are many studies that have developed the type of electromagnetic wave used to activate monoclonal antibodies and photosensitizer by using lasers or X-rays instead of regular light sources and the use of optical filters and the extent to which these filters affect the activation process and results. I hope you take these important factors into account in the test.
Response:
Thank you for your valuable comments. We recognize that various electromagnetic waves, such as lasers, X-rays, and optical filters, play important roles in various phototherapies. We agree that exploring these technical elements is vital in future research. Based on your suggestion, we have added information regarding the potential of different light sources and optical systems to enhance therapeutic effects in lines 532-537.
Reviewer 2 Report
Comments and Suggestions for Authors
The review article " Near-infrared photoimmunotherapy in brain tumors - an unexplored frontier" is devoted to the application of PIT for the treatment of brain tumors. PIT became hot field after the approval of Akalux (Cetuximab-IR700DX conjugate, RM-1929) in Japan. The authors often mention this drug as well as its photoactive part IR700DX in the text; even in the abstract, the authors write that PIT contains an antibody and a photosensitizer, necessarily IR700. However, it should be noted that IR700 is not the only option for photoimmunotherapeutic conjugate; there have been attempts to create conjugates of various antibodies with photoactive dyes with a different mechanism of action than IR700DX. The review is rather superficial, with much attention given to unnecessary discussion of targets for which conjugates for PIT have never been created, but other therapeutic options.
The topic of the review is a very complex one, as no antibody conjugates that actually work in primary brain tumors are currently available, no matter what the therapeutic burden is. Only relatively recently has there been evidence that some individual conjugates, such as Enhertu, can have a meaningful therapeutic effect on breast cancer metastases to the brain and CNS, again, it has not been shown that it is the antibodies and not the therapeutic load that penetrates. In the context of such problems, it is difficult to hope that NIR-PIT antibody conjugates will perform better and penetrate better than antibody conjugates with lipophilic cytotoxic drugs. The authors paid very little attention to this problem.
Other drawbacks include the following:
- The authors cite incorrect data on clinical trials of Akalux for FDA approval (line 123-124). The correct trial number is NCT03769506. The abbreviation is also incorrect.
- Little attention has been paid to the photochemical properties of IR700 and other photosensitizers. To what depth does red light penetrate? An approximate interval could be given, as well as therapeutic approaches that would allow the use of light-demanding conjugates on brain cancer patients.
- Table 1 compares, for reasons that are unclear, Acalux, with its unique mechanism of action, and a common, free photoactive dye used in clinical practice. Why did the authors choose to compare these particular drugs? After all, there are quite a few PDT-approved low molecular weight drugs and they are all different. It is also unclear what significance this should have specifically in relation to brain cancer.
- NovoTTF is not a common therapeutic agent for the treatment of brain cancer.
- A lot of the authors focus on describing some hypothetical GBM targets that have nothing to do with PIT. In the review, it would make sense to consider only those targets for which attempts have been made to produce conjugates for PIT, or at least other structurally similar antibody conjugates. Either there should be a justification why these targets are described with all the details, what is their promise for PIT treatment of brain cancer. The description of vaccines and other drugs is redundant.
- No photosensitizer or conjugate structure is provided.
Thus, the review does not reveal the stated topic of how PIT can help in the treatment of brain cancer, what are the current developments in this field, and what advantages they have over other therapeutic options. Thus, the review can be accepted for publication after round of major revision.
Author Response
Reviewer 2
Comment 1
> It should be noted that IR700 is not the only option for photoimmunotherapeutic conjugate; there have been attempts to create conjugates of various antibodies with photoactive dyes with a different mechanism of action than IR700DX.
Response:
Thank you for this suggestion. As you mentioned, there are multiple cancer therapies known as photoimmunotherapy that use photosensitizers and activating light. Regardless of their mechanisms of action, light-based therapies, including therapy using IR700, have been given various names worldwide. The definitions of these names are vague, which is likely to lead to misunderstanding.
We would like to clarify that while there are multiple cancer therapies that use photosensitizers and activating light, the specific mechanism of NIR-PIT using IR700 is unique. IR700 forms aggregates upon near-infrared light irradiation, resulting in rapid and specific cell membrane damage, a mechanism that is distinct from other photosensitizers that typically work through the generation of reactive oxygen species.
At least in Japan, the therapy using IR700 is called near-infrared photoimmunotherapy (NIR-PIT). Therefore, in this review, we refer to the photoimmunotherapy using IR700 that forms aggregates upon near-infrared light irradiation, resulting in cell membrane damage, as “NIR-PIT.”
In this state of confusion, failing to explain this definition first could lead to further misunderstanding, so we have included an explanation of it in the Introduction (lines 61-67).
In the development of photoimmunotherapy, in which near-infrared light induces the formation of aggregates that damage cell membranes, research is advancing to synthesize fluorescent dyes other than IR700. However, no dyes with efficiencies greater than IR700 have been reported thus far.
Comment 2
>The review is rather superficial, with much attention given to unnecessary discussion of targets for which conjugates for PIT have never been created, but other therapeutic options.
Response:
Thank you for your insightful comment. You are right that NIR-PIT is not currently used for the treatment of brain tumors, and we acknowledge that our review discusses several potential targets for which NIR-PIT conjugates have not yet been publicly reported. While this may appear superficial, we believe that there is significant value in comprehensively analyzing potential targetable proteins across different brain tumor types. Additionally, you mentioned that no conjugates have been created thus far; however, we have already developed some of them that we have not yet published.
As you pointed out, the section subtitle "Near-infrared photoimmunotherapy for brain tumors" is an exaggeration. Therefore, we have revised it to “Therapeutic Potential of NIR-PIT: Consideration of Targetable Surface Antigens Based on Brain Tumor Type. "
Comment 3
>The topic of the review is a very complex one, as no antibody conjugates that actually work in primary brain tumors are currently available, no matter what the therapeutic burden is. Only relatively recently has there been evidence that some individual conjugates, such as Enhertu, can have a meaningful therapeutic effect on breast cancer metastases to the brain and CNS, again, it has not been shown that it is the antibodies and not the therapeutic load that penetrates. In the context of such problems, it is difficult to hope that NIR-PIT antibody conjugates will perform better and penetrate better than antibody conjugates with lipophilic cytotoxic drugs. The authors paid very little attention to this problem.
Response:
Thank you for raising these critical points regarding the challenges of antibody penetration into brain tumors. We acknowledge that the blood-brain barrier (BBB) presents a significant obstacle for antibody-based therapies, including NIR-PIT.
As you have pointed out, there are currently no antibody drugs for brain tumors that show the same level of effectiveness as for extracranial cancers. However, this does not mean that NIR-PIT cannot be used for brain tumors. It is important to note that NIR-PIT does not necessarily require superior antibody penetration compared with antibody-drug conjugates with lipophilic cytotoxic agents. The mechanisms of action differ substantially, which may allow NIR-PIT to be effective, even with limited penetration in certain contexts.
In NIR-PIT, the therapeutic effect of the antibody does not rely on intracellular signaling pathways; rather, it only needs to bind to the cell surface, where light activation can physically destroy the cell membrane. Therefore, the pharmacological activity of these antibodies is less critical. Furthermore, if different epitopes are targeted, antibody drugs can be used in combination with NIR-PIT.
Comment 4
>The authors cite incorrect data on clinical trials of Akalux for FDA approval (lines 123-124). The correct trial number is NCT03769506. The abbreviation is also incorrect.
Response:
Thank you for pointing this out. We have corrected it to “ClinicalTrials.gov NCT Identifier: NCT03769506, Study Number: ASP-1929-301” (lines133-134).
Comment 5
> Little attention has been paid to the photochemical properties of IR700 and other photosensitizers.
Response:
Thank you for this suggestion. We have included a more detailed description of the photochemical properties of IR700 (lines 125-130), a specific photosensitizer relevant to NIR-PIT. Additionally, we provide further information about IR700's photochemical characteristics and its advantages over other photosensitizers in lines 146-155. As our manuscript focuses specifically on the application of IR700-based NIR-PIT for brain tumors, we chose not to expand the properties of other photosensitizers to maintain a clear focus on the clinically relevant approach for this particular therapeutic context.
Comment 6
>To what depth does red light penetrate? An approximate interval could be given, as well as therapeutic approaches that would allow the use of light-demanding conjugates on brain cancer patients.
Response:
Thank you for your comment. The wavelengths used in clinical practice for PDT and NIR-PIT for brain tumors are 664 and 690 nm, respectively. As shown in Table 1, near-infrared light (690 nm) penetrates approximately 2 cm in the tissue, whereas red light (664 nm) penetrates less than 10 mm.
In the initial version of the manuscript, 664 nm light was described as red light, while 690 nm light was referred to as near-infrared light. However, this terminology is misleading and can lead to confusion. Currently, the 690 nm light used in NIR-PIT is typically described as near-infrared light in the literature, so this terminology has been retained; the term red light has been omitted, and the wavelengths have been clarified where necessary.
For light delivery approaches, we have added a new section "7. NIR-PIT for Brain Tumors describes techniques for both shallow lesions (frontal diffuser) and deep lesions (cylindrical diffuser insertion). We also discussed potential approaches to enhance the penetration, including longer wavelengths, optical filters, pulse irradiation, and X-ray-sensitive materials.
Comment 7
>Table 1 compares, for reasons that are unclear, Acalux, with its unique mechanism of action, and a common, free photoactive dye used in clinical practice. Why did the authors choose to compare these particular drugs?
After all, there are quite a few PDT-approved low molecular weight drugs and they are all different. It is also unclear what significance this should have specifically in relation to brain cancer.
Response:
Thank you for your comment. As you have noted, talaporfin sodium is not the only photosensitizer available for PDT. However, talaporfin sodium-based PDT has been clinically approved for glioblastoma treatment in Japan since 2013. Similarly, NIR-PIT has been approved only in Japan and is currently in clinical use. For these reasons, we compared Acalux (NIR-PIT) with PDT using talaporfin sodium.
When evaluating the potential of NIR-PIT for brain tumor applications, comparing it to an established light-activated therapy already used clinically for brain tumors provides the most relevant clinical context. This comparison highlights key differences in the mechanism of action, tissue penetration depth, and tumor selectivity, which are particularly critical for brain tumor treatment, where the preservation of normal tissue function is paramount.
Comment 8
> NovoTTF is not a common therapeutic agent for the treatment of brain cancer.
Based on your suggestion, we have removed the term “NovoTTF” from the manuscript. As you well know, the EF-14 trial is a randomized Phase 3 trial combining TTFields with radiation and temozolomide treatment, and, along with the Stupp temozolomide trial, is one of the few glioblastoma trials that can be considered as high level of evidence. TTFields is thus approved for the treatment of newly diagnosed glioblastomas in the US, many European countries, as well as Japan. It is also recommended in the NCCN guidelines as Category 1 for the treatment of newly diagnosed glioblastomas. The fact that it is a relatively expensive treatment may hinder its dissemination in Western countries.
Comment 9
> A lot of the authors focus on describing some hypothetical GBM targets that have nothing to do with PIT. In the review, it would make sense to consider only those targets for which attempts have been made to produce conjugates for PIT, or at least other structurally similar antibody conjugates. Either there should be a justification why these targets are described with all the details, what is their promise for PIT treatment of brain cancer. The description of vaccines and other drugs is redundant.
Response:
Thank you for your comment. As you know, NIR-PIT has not yet been performed on brain tumors, and complete treatment efficacy has not been demonstrated even in animal models. While our discussion of NIR-PIT for brain tumors remains largely speculative, we still believe that identifying candidate receptors for NIR-PIT in various brain tumor types is beneficial. The receptors we describe are promising candidates for NIR-PIT targeting because other approaches have already validated them as targets. Discussing these potential targets helps bridge the gap between the current knowledge and future applications. It also helps researchers develop NIR-PIT conjugates for brain tumors by highlighting targets with proven clinical relevance.
In addition, we believe that there are limitations to focusing only on currently developed conjugates. One of our main aims is to investigate and discuss the future possibilities in this field.
Comment 10
> No photosensitizer or conjugate structure is provided.
Response:
The structure of the IR700 is shown in Figure 1. We have included a description of the structure of the conjugate in lines 116-119.
Comment 11
>The review does not reveal the stated topic of how PIT can help in the treatment of brain cancer, what are the current developments in this field, and what advantages they have over other therapeutic options. Thus, the review can be accepted for publication after round of major revision.
Response:
Thank you for this suggestion. We have added a new section (7. NIR-PIT for Brain Tumors) that covers possible treatment approaches and advantages. NIR-PIT for brain tumors has only been reported in mouse experiments, which showed tumor reduction and immune response activation. Therefore, there is insufficient information available to discuss the recent advances in detail. Although this is largely speculative, this review attempts to rationally and logically explore the possibilities of this undeveloped field from the current clinical perspective.
Round 2
Reviewer 2 Report
Comments and Suggestions for Authors
The authors have considered the comments and made the necessary changes. The article can be published in its current form.